# Effect of the proximal secondary sphere on the self-assembly of tetrahedral zinc-oxo clusters

Michał Terlecki [1], Iwona Justyniak [2], Michał K. Leszczyński [1,2] & Janusz Lewiński [1,2]✉

Metal-oxo clusters can serve as directional and rigid building units of coordination and noncovalent supramolecular assemblies. Therefore, an in-depth understanding of their multi-faceted chemistry is vital for the development of self-assembled solid-state structures of desired properties. Here we present a comprehensive comparative structural analysis of isostructural benzoate, benzamidate, and new benzamidinate zinc-oxo clusters incorporating the [O,O]-, [O,NH]- and [NH,NH]-anchoring donor centers, respectively. We demonstrated that the NH groups in the proximal secondary coordination sphere are prone to the formation of intermolecular hydrogen bonds, which affects the packing of clusters in the crystal structure. Coordination sphere engineering can lead to the rational design of new catalytic sites and novel molecular building units of supramolecular assemblies.

[1] Faculty of Chemistry, Warsaw University of Technology, Noakowskiego 3, 00-664 Warsaw, Poland. [2] Institute of Physical Chemistry, Polish Academy of Sciences, Kasprzaka 44/52, 01-224 Warsaw, Poland. ✉email: lewin@ch.pw.edu.pl

ntermolecular interactions drive the molecular recognition and self-assembly processes in both chemical and biological systems[1–3], as well as govern the supramolecular architecture of modern functional materials[4–6]. In this view, the utilization of noncovalent-driven self-assembly of molecular units is a convenient and economical way to design and construct targeted solid-state structures that have desired properties[7,8]. A tetrahedral oxo-centered zinc carboxylate $Zn_4O(CO_2)_6$ unit, as a prototypical node, has been particularly reticulated with various organic linkers into extended metal-organic frameworks (MOFs), including the archetypical MOF-5[9,10]. This type of metal-oxo clusters not only serve as directional and rigid building units but also can act as effective precursors of zinc oxide nanostructures[11], gas absorption sites[12–14], and efficient catalyst in transesterification of esters[15–18], esterification of secondary amides[19], acetylation and de-acetylation of carbohydrates[20], direct conversion of esters, lactones, and carboxylic acids to oxazolines[16,21], and $CO_2$ fixation to cyclic carbonates[22–25]. Therefore, an in-depth understanding of the multifaceted chemistry of zinc-oxo clusters supported by various types of organic ligands is continually vital for the development of various Zn-based functional materials. While a number of zinc-oxo clusters incorporating carboxylates[18,26–31], carbamates[32], amidates[10], amidinates[33,34], guanidinates[35,36], pyrrolylketiminates[37], or phosphinates[38,39] were prepared, their reactivity was studied to a lesser extent[17,29,39–42]. We demonstrated that the supporting organic ligands in this type of clusters are prone to exchange under solvent-free conditions, which paved the way for the development of both a new efficient mechanochemical approach for isoreticular MOF (IRMOF) materials[10,43] and a secondary building unit-based mechanochemical approach for drug-loaded MOFs[44]. This was enabled by the control of the acid–base relationship between substituting reagents and synthesis of a new zinc-oxo precursor supported by a benzamidate ligand of relatively high basicity[10]. Oxo-centered zinc carboxylates were also used as a predesigned platform for modeling both their catalytic activity[17,18,21,23] and prototypical Zn-MOFs' reactivity toward water and solvent molecules[14,29,45]. These investigations showed that the zinc centers of the $\{Zn_4O\}^{6+}$ core can easily extend their primary and secondary coordination sphere (hereafter referred as PCS and SCS, respectively) by both donor–acceptor and noncovalent interactions without breaking of the initial Zn–$O_{carboxylate}$ bonds (Fig. 1a). Remarkably, the Zn-coordinated $H_2O$ molecules provided H-bonding sites capable of interacting with donor solvent molecules in the SCS, which led to a unique interface of the $Zn_4O(CO_2)_6$ unit. Furthermore, very recently we investigated the dynamics of PCS and SCS, and the supramolecular structure of a predesigned polyhedral oxo-centered zinc amidinate cluster comprising remote phenyl subunits (Fig. 1b)[46]. Due to the high adaptability of the SCS, this cluster proved to be an excellent model system to study multistep polymorphic phase transitions promoted by supramolecular interactions.

The supramolecular structure of molecular crystals can be tailored by modifications of the organic skeleton or by mixing various molecular building units, and these approaches can tune the microcavities and functionality of the resulting materials[7,8,47–49]. In this vein, simple zinc-oxo carboxylates and amidinates incorporating various substituents in the aromatic backbone in the distal SCS (Fig. 1c) can form diverse types of noncovalent assemblies ranging from structures representing zeolitic topologies to soft porous materials with gated voids or open channels[30,31,33]. For these molecular systems, the character of the organic ligands' backbone in the SCS plays a crucial role in governing noncovalent interactions and self-assembly processes responsible for the packing of molecules. Herein, we turn our attention to the effect of N-bonded hydrogen atoms prone to the

formation of hydrogen bonds in the proximal core environment on the self-assembly of molecular $\{Zn_4O\}^{6+}$-type clusters (Fig. 2). To this aim, we synthesized and characterized a new zinc-oxo complex $[Zn_4O(L^{NN})_6]$ (1) stabilized by benzamidinate ($L^{NN}$) ligands featuring the two anchoring NH groups. Then, we compared the supramolecular structure and self-assembly properties of 1 with isostructural $[Zn_4O(L^{OO})_6]$ and $[Zn_4O(L^{ON})_6]$ clusters stabilized by benzoate ($L^{OO}$) and benzamidate ($L^{ON}$) ligands incorporating the anchoring [O,O] and [O,NH] donor centers, respectively (Fig. 1c). Comprehensive structural analysis of this family of clusters allowed to demonstrate a profound effect of the composition of the proximal SCS of zinc centers on the self-assembly processes of the selected model compounds. To the best of our knowledge, this type of manipulation of the proximal vs distal SCS hardly ever involve molecular crystal engineering.

## Results and discussion

**Synthesis and characterization of $[Zn_4O(L^{NN})_6]$ (1).** The new zinc-oxo complex 1 was obtained by the controlled hydrolysis of a generated in situ ethylzinc derivative of benzamidine ($L^{NN}$-H) in a tetrahydrofuran (THF) solution (Fig. 2). One equivalent of $H_2O$ was added to the reaction mixture containing $ZnEt_2$ and $L^{NN}$-H in a 4:6 molar ratio. After 2 days of storing of the parent solution at −20 °C, big pillar crystals of a 1·5THF solvate (hereafter $1^{LT}$) were isolated and characterized by single-crystal X-ray diffraction (SC-XRD), [1]H nuclear magnetic resonance (NMR), Fourier transform infrared (FTIR) spectroscopy, and elemental analysis (see Supporting information, sections 1–4). Interestingly, the as-received crystal phase $1^{LT}$ was stable up to about 10 °C. Above this temperature, the irreversible phase transition into a new crystal phase $1^{RT}$ occurs affording a polycrystalline material (Fig. 2 and see Supporting information, section 5). The same crystal phase $1^{RT}$ was also easily obtained with almost quantitative yield by crystallization of 1 at ambient temperature. Unfortunately, all attempts to obtain monocrystals of $1^{RT}$ suitable for SC-XRD analysis resulted in fine crystalline suspensions. The structural identity of both solvomorphs was confirmed by [1]H NMR and FTIR spectroscopy (see Supporting information, sections 3 and 4), and the combining elemental and thermogravimetric (TGA) analysis reveals the molecular composition for $1^{RT}$ as a 1·THF solvate (see "Methods" and Supporting information, section 6). Moreover, the TGA data of $1^{RT}$ indicates relatively deep inclusion of the solvent molecules into the host cavity.

SC-XRD analysis of $1^{LT}$ revealed that compound 1 comprises the common tetrahedral $\{Zn_4O\}^{6+}$ core stabilized by six $\mu_2$-bridging monoanionic benzamidinate ligands (see Supporting information, section 2). Interestingly, the anchoring NH groups provide H-donor sites for four THF molecules confined in the SCS of 1 (the O···H distances are about 2.24 and 2.33 Å). In addition, another THF molecule is included within a supramolecular network of $1^{LT}$ with no specific hydrogen interactions (vide infra), and finally, the stoichiometry of the resulting solvate is 1·5THF. The molecular structure of 1 is essentially isostructural with the benzoate $[Zn_4O(L^{OO})_6]$[27] and benzamidate $[Zn_4O(L^{ON})_6]$[10] zinc-oxo analogs featuring susceptible to noncovalent interactions remote aromatic backbones arranged in an octahedral geometry around the core. Thus, these types of zinc-oxo clusters seem ideal tectons for supramolecular structure propagations. Below, we analyze their supramolecular architectures in detail for a more in-depth understanding of the influence of the proximal SCS of zinc-oxo clusters on their self-assembly. Despite the similarities in the molecular structure, all these complexes form different supramolecular assemblies. To better visualize the effect of the character of the proximal SCS, and particularly the subsequent introduction of NH groups to the

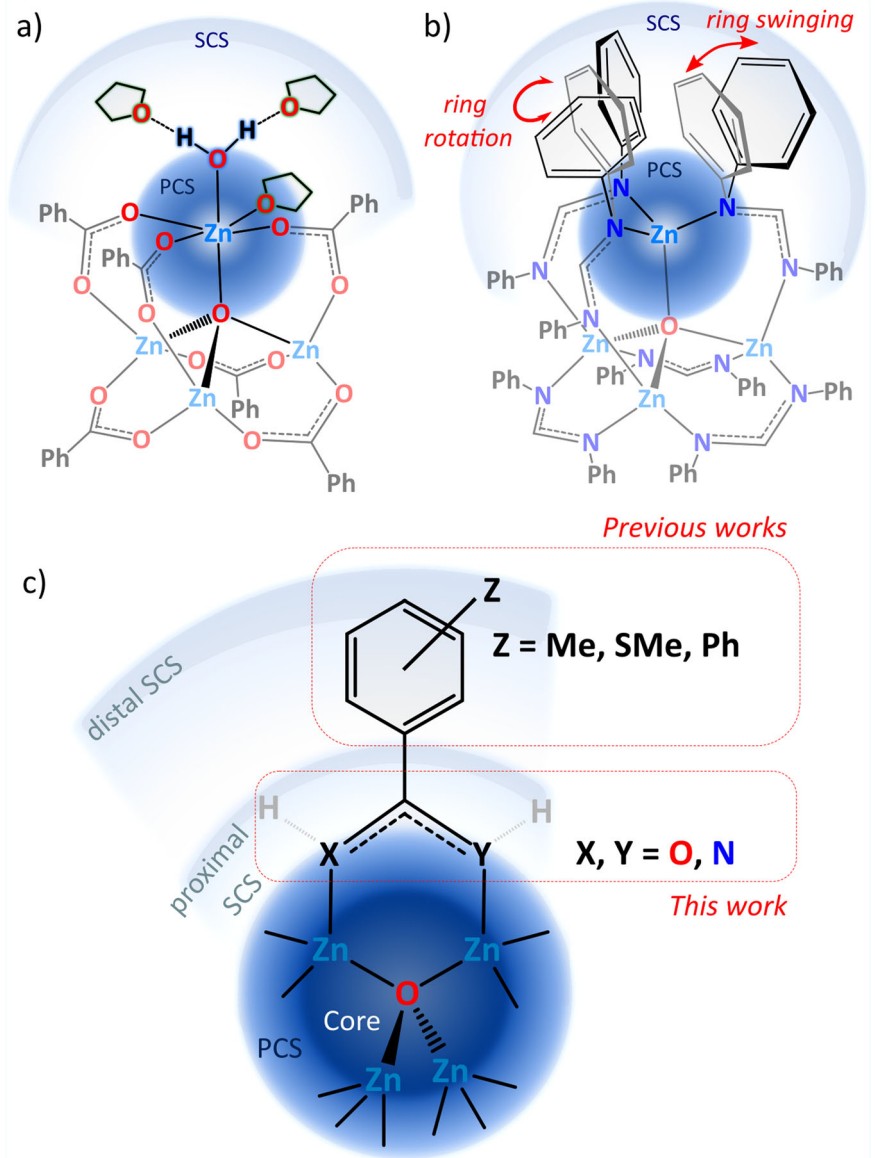

**Fig. 1 Multifaceted chemistry of zinc-oxo clusters stabilized by various organic ligands. a** Unique interface of the hydrated zinc-oxo carboxylate, **b** dynamics of the SCS of a predesigned zinc-oxo amidinate, and **c** schematic representation of the PCS and SCS engineering of model zinc-oxo clusters stabilized by various bidentate ligands with aromatic subunits.

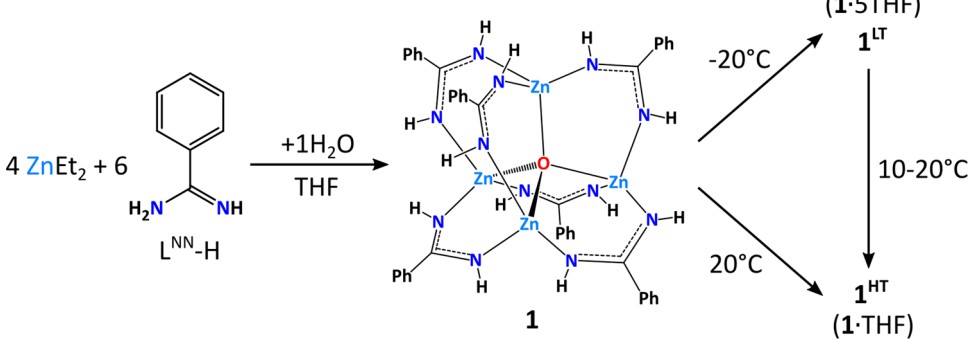

**Fig. 2 Synthesis of zinc-oxo cluster 1.** Schematic representation of the synthesis of zinc-oxo cluster **1** and their transformations to the crystal phases **1$^{LT}$** and **1$^{RT}$**.

anchoring shell, the analysis is presented in the following order starting from the $[Zn_4O(L^{OO})_6]$ benzoate to benzamide $[Zn_4O(L^{ON})_6]$ and benzamidinate $[Zn_4O(L^{NN})_6]$ clusters comprising the [O,O]-, [O,NH]- and [NH,NH]-anchoring groups, respectively. We also note that all these clusters were isolated from the parent THF solutions and the donor solvent affected their supramolecular structure to varying degrees.

**Effect of NH groups in the proximal SCS on the self-assembly of zinc-oxo clusters**. The supramolecular structure of the cluster $[Zn_4O(L^{OO})_6]$ is governed only through multiple cooperative CH–π and π–π interactions in the distal SCS as the carboxylate-anchoring groups comprise only H-acceptor O centers (Fig. 3a). The benzoic zinc-oxo molecules self-assemble producing an extended regular 3D network, which topologically resembles a natural porous zeolite analcite with open-gated voids that occupy about 39.3% of the unit-cell volume (Figs. 3d and 4a)[27,30]. Nevertheless, despite the observed open supramolecular framework architecture, very small aperture sizes make the voids free of solvent THF molecules and the material does not show any significant gas uptake. It should also be noted that an extension of the ligand backbone by various substituents in the remote aromatic rings strongly affects the supramolecular arrangement of zinc-oxo carboxylates, which is an effect of the low specificity of CH–π and π–π interactions[30].

While the benzamide analog $[Zn_4O(L^{ON})_6]$ served us previously as an efficient precursor in the mechanochemical synthesis of IRMOF materials[10], its supramolecular structure was not analyzed in detail. The benzamide zinc-oxo cluster comprises six anchoring NH groups susceptible to both the formation of hydrogen bonds with donor solvent molecules and the hydrogen-bonding-mediated molecular self-assembly. Indeed, $[Zn_4O(L^{ON})_6]$ molecules comprising both the electronegative O atom and the NH group in the PCS acts as cogwheels, which interlock via pairs of complementary intermolecular NH⋯O hydrogen bonds in the proximal SCS (Fig. 3b). The crystal structure is almost completely dominated by molecular H-bonded aggregations and simultaneously supported by an extensive array of weaker π–π interactions between the aromatic rings in the distal SCS (Fig. 3b). The assembled $[Zn_4O(L^{ON})_6]$ molecules form hexagonal supramolecular 2D layers with the $\bar{3}$ symmetry, with AA-type stacking into a honeycomb supramolecular structure (Fig. 3e). Although compound $[Zn_4O(L^{ON})_6]$ crystallizes as a solvate with 1.5 molecules of THF and that THF molecules are not involved in the direct cluster solvation by N–H⋯O hydrogen bonds, they are encapsulated in closed cavities between the supramolecular 2D layers (Fig. 4b).

The benzamidinate cluster **1** comprises only N–H donor groups in the PCS and it crystallizes from a THF solution at −20 °C as a solvate **1**·5THF. Each zinc-oxo cluster forms hydrogen bonds with four THF molecules (Fig. S3c) and one THF molecule is included within a supramolecular network of $\mathbf{1^{LT}}$ with no specific hydrogen interactions (Fig. 3h). Consequently, the resulting solvate may be formulated as [**1**·4THF]·THF. Lack of the hydrogen bond acceptor O centers in the PCS excludes the formation of complementary intermolecular NH⋯O interactions and thus favors the direct cluster solvation by the H-bonded THF. The H-bonded THF molecules confined in the SCS of **1** efficiently hinder access to the proximal SCS interior and affect an interlocking mechanism of the zinc-oxo cogwheels (Fig. 3c). The solvated **1**·THF molecules self-assemble via cooperative CH–π and π–π noncovalent interactions mediated by the aromatic rings in the distal SCS affording a honeycomb supramolecular structure composed of AB-type stacked trigonal layers (Figs. 3f and 4c). In this view, the benzamidinate zinc-oxo clusters act as cogwheels with shallower gear elements compared to that in benzamide analogs. As a result, the direct THF solvation enforces an extension of the honeycomb supramolecular structure by about 24% (Fig. 3g), and leads to the formation of 1D open channels along the crystallographic *c* direction in the $\mathbf{1^{LT}}$ supramolecular framework, occupying about 11.0% of the unit-cell volume. These channels are tightly filled by non-H-bonded THF molecules (Figs. 3h and 4c). Notably, activation of the material and the utilization of its porous properties are significantly impeded by the thermal instability of $\mathbf{1^{LT}}$ (vide supra), which causes a negligible $N_2$ uptake (Fig. S11). The observed phase transition to the $\mathbf{1^{RT}}$ likely occurs due to thermal dissociation of the NH⋯THF hydrogen bonds, which trigger the rearrangement of the honeycomb structure. The new supramolecular structure of $\mathbf{1^{RT}}$ comprises only one THF molecule, tightly encapsulated within the host network (see Supporting information, section 6).

## Conclusion

The synthesis of targeted solid-state structures of desired properties through an understanding and control of intermolecular interactions in the crystal is one of the major challenges of crystal engineering. Surprisingly, crystal engineering endeavors hardly ever involve the manipulation of the proximal vs distal SCS. Filling that gap, we turn our attention to a series of the isostructural tetrahedral zinc-oxo clusters featuring the $Zn_4(\mu_4\text{-}O)$ core and incorporating [O,O]-, [O,NH]- and [NH,NH]-anchoring donor centers. The comprehensive analysis of their molecular and crystal structures demonstrated the profound influence of the character of the PCS and SCS on the solvation processes (which can be mediated by both donor–acceptor and hydrogen bonding interactions) as well as the dimensionality and topology of their supramolecular solid-state structures. In this vein, the presented investigation along with our previous studies on the reactivity of oxo-centered zinc carboxylates toward water and donor solvents[29] should pave the way for a more efficient coordination sphere engineering of new catalytic sites and novel molecular building units of supramolecular functional assemblies.

## Methods

All manipulations were conducted under a nitrogen atmosphere using standard Schlenk techniques. All reagents were purchased from commercial vendors: benzamidine ($L^{NN}$-H) (Sigma-Aldrich), $ZnEt_2$ (ABCR). Solvents were purified and dried using MBraun Solvent Purification System (SPS). The $^1H$ and $^{13}C$ NMR spectra were acquired on Varian Mercury (400 MHz) spectrometer. FTIR spectra were recorded on a Bruker-Tensor II spectrometer. Powder X-ray diffraction (PXRD) measurements were performed using a PANalytical Empyrean diffractometer equipped with Ni-filtered Cu Kα radiation (40 kV, 40 mA). The sample for the PXRD analysis was sealed between two layers of Kapton foil and measured in transmission geometry. Elemental analyses were performed on an Elementar VarioMicro Cube analyzer. TGA-differential scanning calorimetry (DSC) analyses were performed under argon with a heating rate of 5 °C min$^{-1}$ using a TA Instruments Q600 apparatus. Volumetric $N_2$ sorption studies were undertaken using a Micromeritics ASAP 2020 system. Approximately 150 mg of $\mathbf{1^{LT}}$ and dried under vacuum at −10 °C for 5 h. Helium was used for the free space determination after sorption analysis. Adsorption isotherms were measured at 77 K in liquid nitrogen.

**Synthesis of $\mathbf{1^{LT}}$**. $ZnEt_2$ (2 M in hexane, 0.5 ml, 1 mmol) was added to a solution of $L^{NN}$-H (180 mg, 1.5 mmol) in 5 ml of THF at −78 °C. The reaction mixture was allowed to warm to room temperature and stirred overnight. Next, the solution was cooled to −20 °C and $H_2O$ was added (1 M in THF, 0.25 ml, 0.25 mmol). The reaction was placed in the freezer (−20 °C) and after 48 h big pillar colorless crystals of $\mathbf{1^{LT}}$ were isolated (yield 197 mg, 64%). $^1H$ NMR ($d_8$-THF, 400 MHz): δ [ppm] = 7.59 (m, 12H, Ar), 7.28 (m, 18H, Ar), 4.71 (s, 12H, NH); $^{13}C$ NMR ($d_8$-THF, 100 MHz): δ [ppm] = 174.64, 144.79, 129.49, 128.99, 126.86; FTIR (ATR): ν [cm$^{-1}$] = 3366 (m), 3058 (w), 2970(m), 2857 (m), 1593 (s), 1560 (s), 1512 (s), 1477 (s), 1442 (m), 1290 (s), 1061 (s), 1027 (m), 905 (m), 785 (m), 732 (m), 698 (s), 682 (s), 518 (m), and 478 (s). Results of the elemental analysis

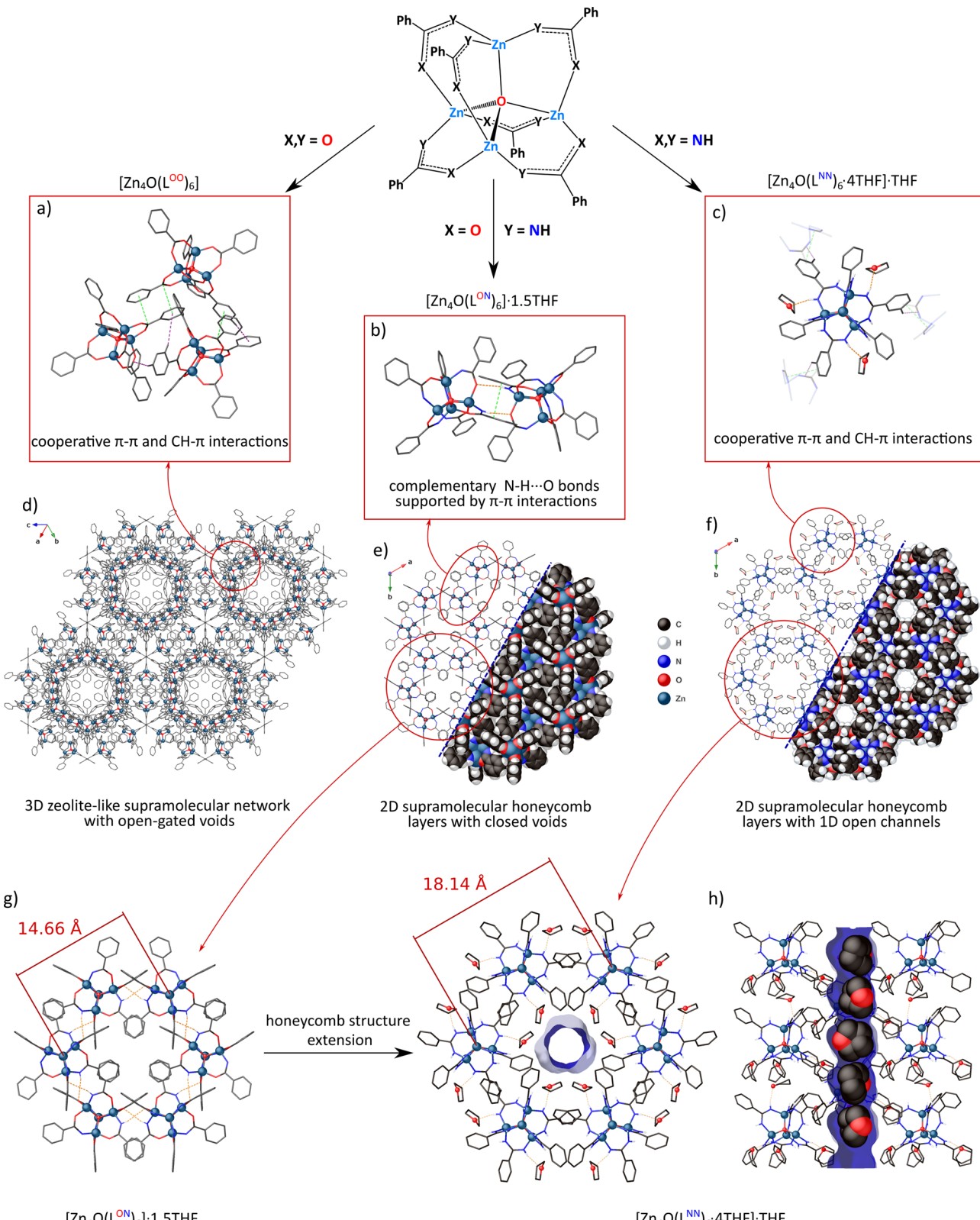

**Fig. 3 Self-assembly of the benzoate, benzamide, and benzamidinate zinc-oxo clusters.** Main intermolecular forces governing the self-assembly processes (**a**–**c**) and supramolecular structures (**d**–**f**) of [Zn₄O(L^OO)₆], [Zn₄O(L^ON)₆]·1.5THF, and [Zn₄O(L^NN)₆·4THF]·THF. **g** Representation of the honeycomb structure extension between benzamidate and benzamidinate derivatives. **h** THF molecules packing in the 1D open channels of [Zn₄O(L^NN)₆·4THF].

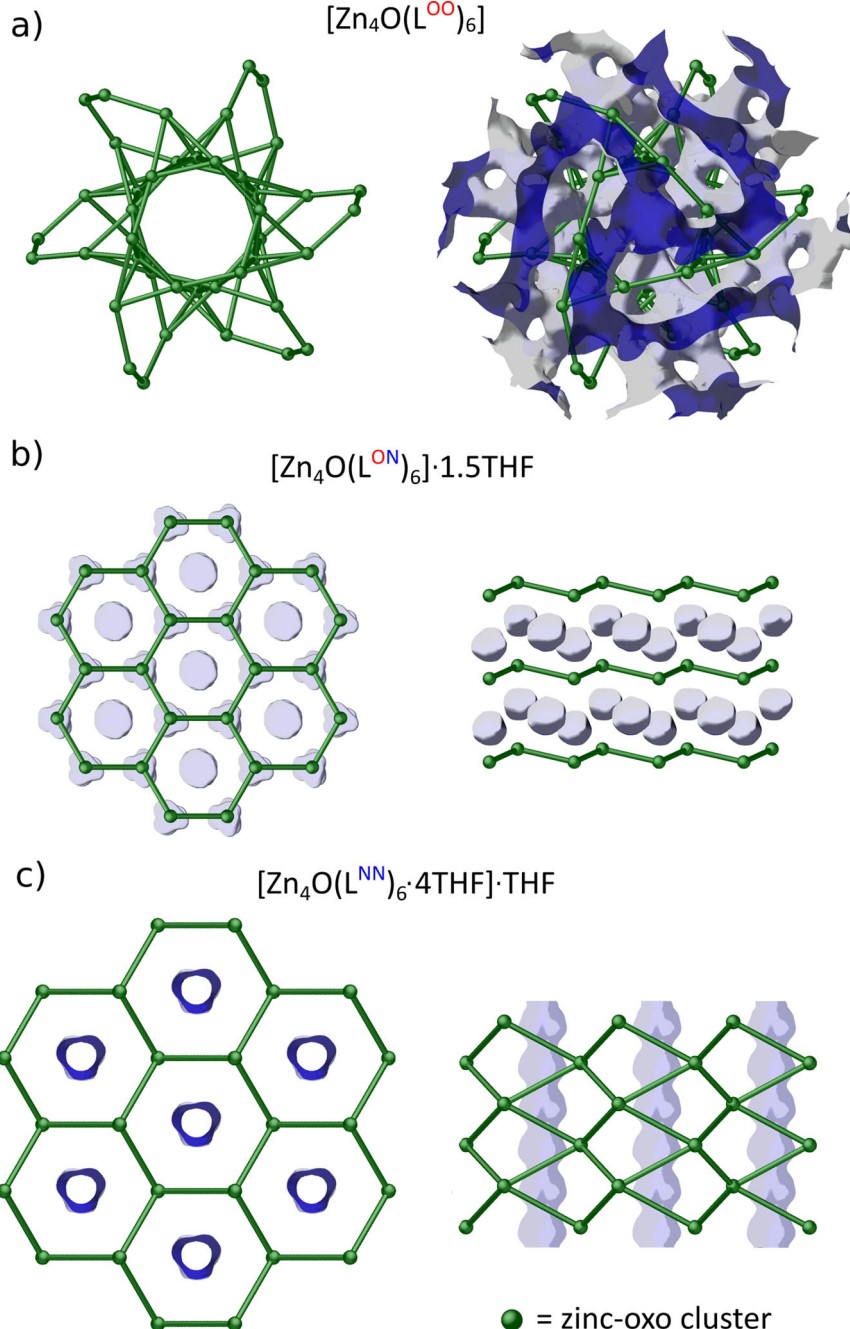

**Fig. 4 Topology of the porous frameworks of the benzoate, benzamidate, and benzamidinate zinc-oxo clusters.** Schematic representation of the supramolecular network formed by of $[Zn_4O(L^{OO})_6]$ (**a**), $[Zn_4O(L^{ON})_6]$ (**b**), and $[Zn_4O(L^{NN})_6 \cdot 4THF]$ (**c**) clusters with solvent-accessible surfaces of voids [blue and gray colors represent the interior and exterior pore surface, respectively]. In the case of benzamidate and benzamidinate derivatives, the voids are filled by THF molecules.

may vary due to various content of the THF molecules in the sample, calcd (%) for $1 \cdot 3.8THF$ ($C_{57.2}H_{72.4}Zn_4O_{4.8}N_{12}$): C 54.25, H 5.76, N 13.27; found: C 54.13, H 5.80, N 13.18.

**Synthesis of $1^{RT}$.** ZnEt$_2$ (2 M in hexane, 0.5 ml, 1 mmol) was added to a solution of $L^{NN}$-H (180 mg, 1.5 mmol) in 5 ml of THF at −78 °C. The reaction mixture was allowed to warm to room temperature and stirred overnight. Next, the stoichiometric amount of H$_2$O was added (1 M in THF, 0.25 ml, 0.25 mmol), and the reaction was stirred overnight. Then, the solvent was removed under vacuum and $1^{RT}$ was collected as a white powder in almost quantitative yield. $^1$H NMR ($d_8$-THF, 400 MHz): δ [ppm] = 7.59 (m, 12H, Ar), 7.28 (m, 18H, Ar), 4.71 (s, 12H, NH); $^{13}$C NMR ($d_8$-THF, 100 MHz): δ [ppm] = 174.66, 144.79, 129.49, 128.99, 126.86; FTIR (ATR): ν [cm$^{-1}$] = 3366 (m), 3058 (w), 2970(w), 2857 (w), 1593 (s), 1560 (s), 1512 (s), 1477 (s), 1442 (m), 1290 (s), 1061 (m), 1027 (m), 905 (w), 785

(m), 732 (m), 698 (s), 682 (s), 518 (m), 478 (s). Results of the elemental analysis may vary due to various content of the THF molecules in the sample, calcd (%) for $1 \cdot 0.855THF$ ($C_{45.44}H_{48.88}Zn_4O_{1.86}N_{12}$): C 51.76, H 4.67, N 15.95; found: C 51.29, H 4.67, N 15.52.

**Crystallographic measurements for $1^{LT}$.** The crystals were selected under Paratone-N oil, mounted on the nylon loop, and positioned in the cold stream on the diffractometer. The X-ray data were collected at 100(2) K on a Nonius KappaCCD diffractometer using MoKα radiation (λ = 0.71073 Å). The data were processed with *DENZO* and *SCALEPACK* (HKL2000 package)[50]. The structures were solved by direct methods using the SHELXS-97 program and were refined by full-matrix least-squares on F$^2$ using the program SHELXL[51]. All non-hydrogen atoms were refined with anisotropic displacement parameters. Hydrogen atoms

were added to the structure model at geometrically idealized coordinates and refined as riding atoms.

## Data availability

The authors declare that the data supporting the findings of this study are available within the paper and its Supplementary information files. Crystallographic data (excluding structure factors) for the structure $1^{LT}$ reported in this paper have been deposited at the Cambridge Crystallographic Data Center under deposition number CCDC: 2072061 (Supplementary Data 1). These data can be obtained free of charge from The Cambridge Crystallographic Data Center via www.ccdc.cam.ac.uk/data_request/cif.

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

## Acknowledgements

We gratefully acknowledge the National Science Center, Poland (Grant OPUS 2017/25/B/ST5/02484) for financial support.

## Author contributions

M.T. planned and carried out the experiments, analyzed most of the experimental data, and also co-wrote the paper. I.J. performed the X-ray single-crystal diffraction analysis. M.K.L. performed the PXRD measurements and assisted in writing the manuscript. J.L. supervised the work and wrote the manuscript.

## Competing interests

The authors declare no competing interests.
