## [Peer Review File · Communications Chemistry]

Reviewers' comments:

Reviewer #1 (Remarks to the Author):

This contribution from Lewiński and coworkers describes the synthetic and structural features of an isostructural family of zinc-oxo clusters of the type $[Zn_4O(L)_6]$ where L contains either O and/or N donor atoms. The effect on the supramolecular structure of the gradual introduction of the NH moiety is investigated and some interesting effects are observed which may prove useful in other systems. The work will be interesting to the coordination chemistry community, and sufficient detail has been added for others to repeat the work. The manuscript is also well written. I am in favour of publication. Note references 20 and 34 are the same. The elemental analysis is 1RT is not great.

Reviewer #2 (Remarks to the Author):

Lewinski et al. Describe the synthesis of a new benzamidinate zinc-oxo cluster and compared its supramolecular structure and self-assembly properties with isostructural benzoate and benzamidate derivatives. In general, the chemistry seems to have been performed in a proper manner and the results are interesting. However, I do not see the urgency to publish this work as a communication. The results are nice, but they are distant to be comprehensive. And sure enough I do agree that these results may serve for the future rational design of similar species with potential interesting properties, but at this point this asseveration is premature. Hence, besides the fact that the MS requires a proper revision of the English grammar and based on the above I can not support publication of this MS in Communications Chemistry – Nature. I believe that this chemistry reformatted as a full paper may be suitable for instance for Dalton Transactions.

Reviewer #3 (Remarks to the Author):

see file

Effect of the Proximal Secondary Sphere on the Self-Assembly of Tetrahedral Zinc-Oxo Clusters

Michał Terlecki, Iwona Justyniak, Michał Leszczyński, Janusz Lewiński

In the title manuscript, Lewiński and co-workers present the synthesis and characterization of a new zinc-oxo cluster stabilized by a benzamidinate ligand, $[L^{NN}]^-$. While the zinc-oxo cluster may be synthesized at both $-78\text{ }^\circ\text{C}$ and room temperature, the products of which being referred to as clusters $\mathbf{1}^{LT}$ and $\mathbf{1}^{RT}$, respectively, an irreversible phase transition occurs at $\sim 10\text{ }^\circ\text{C}$, rendering characterization of $\mathbf{1}^{RT}$ by single crystal X-ray diffraction unattainable. Nevertheless, the collected spectroscopic data is consistent between the two, indicating that both are the same compound. During this study, Lewiński and co-workers compare the structural data obtained from single crystal X-ray diffraction for $\mathbf{1}^{LT}$ with the already reported benzoate and benzamidate analogues, $[\text{Zn}_4\text{O}(\text{L}^{\text{OO}})_6]$ and $[\text{Zn}_4\text{O}(\text{L}^{\text{NO}})_6]$, respectively.

The results reported in the title manuscript are quite slight and highly specific, particularly the rather limited new work being described. A significant weakness of the work is that there is no impact or application consequence to changing the ligand/coordination sphere interactions in this way. It would be a much stronger body of work if there were a reason for the change in ligand and some consequences from it. As it stands, this work appears to be a fairly standard but well-conducted piece of inorganic chemistry and should be suitable for publication but it's a pity as this work could be much stronger if publication were delayed and applications properly explored. Overall, given the scope of the journal, the work appears suitable for publication but there are some aspects that must be corrected.

The following suggested revisions must be addressed prior to final acceptance for publication:

- ^1H and $^{13}\text{C}\{^1\text{H}\}$ NMR data and IR spectroscopic data are provided for $\mathbf{1}^{RT}$ and $\mathbf{1}^{LT}$ but no assignments have been made in the Experimental section. The required 2D-NMR experiments should be used to accurately assign the ^1H and ^{13}C resonances observed.
- The results from elemental analysis fall well outside the acceptable range for publication (i.e. more than 1 % for both C and N for $\mathbf{1}^{RT}$ when the acceptable range is 0.3 %) – this issue must be addressed.
- The packing and space-fill model from the single crystal X-ray diffraction data for $\mathbf{1}^{LT}$ depicted in Figure 2 is unfortunately misleading. The authors claim that 1D open channels are formed throughout the lattice, however, upon opening the cif in Mercury and examining the extended packing diagram, disordered lattice solvent (presumably cyclohexane) is found within the 1D channels and has been removed from the Figure (see appended Figure here). This must be corrected in the manuscript.
- Following on from the previous point, the authors stated that gas sorption measurements could not be performed due to thermal instability; could this be due to loss of lattice solvent as opposed to an irreversible phase transition? Were such measurements performed at low temperature? Perhaps if they were, a lack of reactivity could be attributed to the presence of lattice solvent in the potentially accessible voids.
- With respect to the X-ray crystal data, lattice solvent should be indicated in the molecular formula. Furthermore, the disordered molecule of cyclohexane has not been refined and

modelled properly and must be corrected, and an additional section entitled “refine_special_details” that outlines the restraints and methodology used when modelling disorder in the crystal structure must be added into the cif before publication.

- Figure 1c: The reader expects that the proximal core environment will be investigated first followed by L^{NN} bearing different substituents in the aromatic backbone. Is this part of the future work for these systems?
- Results and Discussion – A synthetic scheme should be included in the manuscript for the synthesis of cluster **1**.
- There are many small errors in English language which should be corrected (e.g. missing articles).
- Supporting information: Fig. S4/S5: Label * in NMR spectra as d_8 -THF.
- Supporting information: Fig. S6. FTIR analysis: It would be nice to have the characteristic stretching bands added into the figure as well as in the caption.
- Supporting information: Fig. S7/S8: The labels (a), (b), (c) and so on in the figure caption should be added beside the corresponding powder X-ray diffraction pattern in the figure.

Point-by-point responses to the Reviewers' Comments concerning our submission No COMMSCHEM-21-0128 entitled "Effect of the proximal secondary sphere on the self-assembly of tetrahedral zinc-oxo clusters"

Reviewer #1 (Remarks to the Author):

This contribution from Lewiński and coworkers describes the synthetic and structural features of an isostructural family of zinc-oxo clusters of the type $[Zn_4O(L)_6]$ where L contains either O and/or N donor atoms. The effect on the supramolecular structure of the gradual introduction of the NH moiety is investigated and some interesting effects are observed which may prove useful in other systems. The work will be interesting to the coordination chemistry community, and sufficient detail has been added for others to repeat the work. The manuscript is also well written. I am in favour of publication. Note references 20 and 34 are the same. The elemental analysis is IRT is not great.

Reply: We appreciate that the Reviewer found our work interesting. The elemental analysis for I^{RT} has been reevaluated in the view of new solvate stoichiometry, which was supported by the TGA analysis. The doubled references were corrected.

Reviewer #2 (Remarks to the Author):

Reply: We greatly appreciate the Reviewer #2 thoughtful comments which helped to considerably improve the quality of our manuscript. We trust that most of the comments were addressed accordingly in the revised manuscript. In the following, we give a point-by-point reply to the comments.

Lewinski et al. Describe the synthesis of a new benzamidinate zinc-oxo cluster and compared its supramolecular structure and self-assembly properties with isostructural benzoate and benzamidate derivatives. In general, the chemistry seems to have been performed in a proper manner and the results are interesting.

Reply: We appreciate that the Reviewer found that the experiments were carried out correctly and our work is interesting.

However, I do not see the urgency to publish this work as a communication. The results are nice, but they are distant to be comprehensive.

Reply: We appreciate the Reviewer's statements that the reported results are nice but in turn, we are somewhat puzzled that he/she does not see the urgency to publish this work as a communication.

Polyhedral metal-oxo clusters are of great interest due to tunability of their molecular structures and a vast array of potential applications. At the same time the synthesis of targeted solid-state structures with desired properties through an understanding and control of intermolecular interactions in the crystal is one of the major challenges of crystal engineering. Surprisingly, despite many years of research in the field crystal engineering endeavors hardly ever involve the manipulation of the proximal vs distal secondary coordination sphere (SCS). We selected a series of zinc oxo clusters as a model system and made first attempts to fill this research gap. Thus, taking into account both the importance of the topic and provided our conceptual and experimental contribution, we are convinced that even a communication form seems justified, and the publication will be an inspiration for readers.

And sure enough I do agree that these results may serve for the future rational design of similar species with potential interesting properties, but at this point this asseveration is premature. Hence, besides the fact that the MS requires a proper revision of the English grammar and based on the above I can not support publication of this MS in Communications Chemistry – Nature. I believe that this chemistry reformatted as a full paper may be suitable for instance for Dalton Transactions.

Reply: We again appreciate that the Reviewer fully agrees with our opinion expressed in the main text of our manuscript that the reported results may serve for the future rational design of similar species with potentially interesting properties. However, we cannot agree that this asseveration is premature as such a view is in contradiction with the available literature data. Here, we want to thank the Review for pointing out our attention to this aspect. On this occasion, we added a few additional references in the main text.

A solid foundation for such affirmation has already been substantiated by a number of related publications. For example, zinc-oxo clusters have been widely recognized as directional and rigid building units of MOFs and functional molecular crystals [cf. *Science* **2002**, 295, 469, and *J. Am. Chem. Soc.* **2018**, 140, 15031]. In this view, they have already been used as efficient predesign precursors to developing new synthetic approaches for porous functional materials [cf. *ChemCommun* 2015, 51, 4032 and *Inorg. Chem.* 2018, 57, 13437], and subsequently even more complex host-guest systems like for example drug-loaded MOFs [*Eur. J. Inorg. Chem.* **2020**, 796]. Moreover, zinc-oxo clusters may act as gas absorption [cf. *J. Am. Chem. Soc.* **2009**, 131, 4143] small molecules activation [cf. *Chem. Commun.*, 2012, 48, 7362] or catalysts [cf. newly added references, i.e., ref. 15-25].

Finally, we have carefully edited the manuscript according to the inputs from the two reviewers. We truly hope that the revised manuscript is clear enough to follow. Overall, we believe the presented studies fit well to the communication format and are sufficient for publication. We believe that further delay of publication is unfavorable for this work.

Reviewer #3 (Remarks to the Author):

In the title manuscript, Lewiński and co-workers present the synthesis and characterization of a new zinc-oxo cluster stabilized by a benzamidinate ligand, $[L^{NN}]$ – . While the zinc-oxo cluster may be synthesized at both -78 °C and room temperature, the products of which being referred to as clusters I^{LT} and I^{RT} , respectively, an irreversible phase transition occurs at ~ 10 °C, rendering characterization of I^{RT} by single crystal X-ray diffraction unattainable. Nevertheless, the collected spectroscopic data is consistent between the two, indicating that both are the same compound. During this study, Lewiński and co-workers compare the structural data obtained from single crystal X-ray diffraction for I^{LT} with the already reported benzoate and benzamidate analogues, $[Zn_4O(L^{OO})_6]$ and $[Zn_4O(L^{NO})_6]$, respectively.

Reply: We appreciate that the Reviewer found our experimental work well-conducted.

The results reported in the title manuscript are quite slight and highly specific, particularly the rather limited new work being described.

Reply: We are somewhat puzzled by these statements. As we explained in the response to the Reviewer#2's first major comment, despite many years of research in the field crystal engineering endeavors hardly ever involve the manipulation of the proximal vs distal SCS. As far as we are concerned, we have provided for the first time a comprehensive comparative structural analysis of the isostructural zinc-oxo clusters incorporating the [O,O]-, [O,NH]- and [NH,NH]-anchoring donor centers, and demonstrated the profound effect of the identity of anchoring groups on both their solvation and self-assembly. At this stage of the investigation, we have already painted a rich and insightful picture of the different molecular and supramolecular features concerning the multifaceted chemistry of

polyhedral metal-oxo clusters. Moreover, an in-depth understanding of the impact of the primary (PCS) and secondary coordination spheres (SCS) on their multifaceted chemistry of polyhedral clusters is vital for the development of supramolecular functional materials.

A significant weakness of the work is that there is no impact or application consequence to changing the ligand/coordination sphere interactions in this way. It would be a much stronger body of work if there were a reason for the change in ligand and some consequences from it. As it stands, this work appears to be a fairly standard but well-conducted piece of inorganic chemistry and should be suitable for publication but it's a pity as this work could be much stronger if publication were delayed and applications properly explored.

Reply: There are a huge number of examples that even curiosity-driven basic research has brought truly revolutionary practical solutions. Basic research not only radically alters our deep understanding of the structure-property relationships, but it also leads to new tools that spread throughout practical chemistry and materials science. In the main text of our manuscript, one can find the following statements: "The comprehensive structural analysis demonstrated a profound influence of the compositional alteration of the proximal SCS on the self-assembly of this family of prototypical zinc-oxo clusters and the dimensionality and topology of supramolecular solid-state structures. In this vein, the presented studies along with our previous investigations on the reactivity of oxo-centered zinc carboxylates toward water and donor solvents should pave the way for a more rational coordination sphere engineering of new catalytic sites and novel molecular building units of supramolecular functional assemblies."

We have no doubts that our studies provide a strong foundation and inspiration for further application-driven studies and further delay in publication would be highly disadvantageous for chemists and materials scientists.

In turn, the impact or application consequences to changing the ligand/coordination sphere interactions have already been substantiated by a number of earlier publications (vide supra the response to the Reviewer#2's second major remark). For example, our previous studies on the isostructural tetrahedral zinc-oxo clusters featuring the $Zn_4(\mu_4-O)$ core demonstrated the character of incorporated ligand determines the transformation of zinc-oxo clusters to IRMOFs using mechanochemical synthesis.

Overall, given the scope of the journal, the work appears suitable for publication but there are some aspects that must be corrected. The following suggested revisions must be addressed prior to final acceptance for publication:

Reply: We appreciate that the Reviewer found our manuscript suitable for publication after few corrections. We have revised the manuscript according to the suggestions. We would like to thank the Reviewer for careful reading and valuable comments that improved the quality of this work.

- 1H and $^{13}C\{^1H\}$ NMR data and IR spectroscopic data are provided for 1^{RT} and 1^{LT} but no assignments have been made in the Experimental section. The required 2D-NMR experiments should be used to accurately assign the 1H and ^{13}C resonances observed.

Reply: The NMR and FTIR spectra were described in detail in the SI, where assignments of signals were provided. Moreover, we added appropriate schemes illustrating the signal assignments in Fig. S4 and S5 and marked important bands in the IR spectrum at Fig. S6. This should improve the clarity of the data analysis. Assigning the signals seems quite simple in this case and we don't see any reason for making 2D-NMR experiments that wouldn't add much content.

- The results from elemental analysis fall well outside the acceptable range for publication (i.e. more than 1 % for both C and N for 1^{RT} when the acceptable range is 0.3 %) – this issue must be addressed.

C 50.83, H 4.27, N 16.93; found: C 50.11, H 4.59, N 16.22

Reply: The reported crystal phases are in the form of solvates, which may differ in THF content. In this view, the appropriate comments have been added in the Methods section. Initially, the

composition of both phases was poorly matched to the experimental results, which resulted in some discrepancies. The solvates stoichiometries were recalculated. Moreover, for **1^{RT}** we additionally repeated the elemental analysis on a new sample and the estimated solvate composition was additionally confirmed by the TGA experiment (calc. THF content 5.88%, exp. 5.84%). Elemental analysis is now: **1^{LT}** as **1·3.8THF** (calcd (%): C 54.25, H 5.76, N 13.27; found: C 54.13, H 5.80, N 13.18); **1^{LT}** as **1·0.86THF** (calcd (%): C 51.76, H 4.67, N 15.95; found: C 51.29, H 4.67, N 15.52).

*- The packing and space-fill model from the single crystal X-ray diffraction data for **1^{LT}** depicted in Figure 2 is unfortunately misleading. The authors claim that 1D open channels are formed throughout the lattice, however, upon opening the cif in Mercury and examining the extended packing diagram, disordered lattice solvent (presumably cyclohexane) is found within the 1D channels and has been removed from the Figure (see appended Figure here). This must be corrected in the manuscript.*

Reply: Thank you for your attention in all these matters. Voids in porous supramolecular networks often contain post-synthetic residuals. Omitting them on packing models is a common procedure for the visualization of potential porous properties of materials. Nevertheless, we have significantly modified both Fig. 2 (now Fig. 3) and the supramolecular structure description. We hope it is clear now.

- Following on from the previous point, the authors stated that gas sorption measurements could not be performed due to thermal instability; could this be due to loss of lattice solvent as opposed to an irreversible phase transition? Were such measurements performed at low temperature? Perhaps if they were, a lack of reactivity could be attributed to the presence of lattice solvent in the potentially accessible voids.

Reply: As we mentioned above, the voids in porous networks commonly contain post-synthetic residuals. Upon the removal of these guest molecules, the material may retain its porosity, deformed, or completely collapse. The last behavior is common for noncovalently-bonded networks. We demonstrated that **1^{LT}** is stable during drying under vacuum at low temperatures, but TGA experiment proved that this treatment is insufficient to remove all THF molecules. To achieve this, we had to raise temperature of the activation process, which caused the irreversible phase transition to a new phase **1^{RT}**. Thus, the gas sorption measurement was not initially done due to the inability to properly activate the sample caused by its thermal instability. Nevertheless, in the meantime, we performed this experiment on a sample dried under vacuum, which as predicted shows negligible N₂ uptake at 77K. We have properly modified the description in the manuscript and added adsorption isotherm to the SI.

- With respect to the X-ray crystal data, lattice solvent should be indicated in the molecular formula. Furthermore, the disordered molecule of cyclohexane has not been refined and modelled properly and must be corrected, and an additional section entitled "refine_special_details" that outlines the restraints and methodology used when modelling disorder in the crystal structure must be added into the cif before publication.

Reply: We are grateful for this comment. We took a closer look at the X-ray crystal structure refinement and were able to model THF molecules; they are in disorder on 3-fold symmetry elements. The CIF file has been modified and redeposited in the CCDC database. Accordingly, a description of the supramolecular structure was significantly modified. The molecular formula is now **1·5THF**.

- Figure 1c: The reader expects that the proximal core environment will be investigated first followed by LNN bearing different substituents in the aromatic backbone. Is this part of the future work for these systems?

Reply: The influence of various substituents in the aromatic backbone in the distal SCS on self-organization was previously demonstrated for a series of zinc-oxo carboxylates [*Inorg. Chem.* **2012**, *51*, 7410]. Of course, investigations on the similar effects in the amide systems with self-assembly dominated by intermolecular NH \cdots O interactions in the proximal SCS and the amidinate derivatives prone for direct solvation by THF will be a natural extension of this type of study. We believe that only a comprehensive understanding of the conjugated effects in the proximal and distal SCS gives a more-

in depth understanding of the self-assembly properties of polyhedral metal-oxo clusters and allows for rational manipulation of their supramolecular structure, and thus leading to the tunability of their functionality. These investigations are ongoing in our group and will be published in due course.

- *Results and Discussion – A synthetic scheme should be included in the manuscript for the synthesis of cluster 1.*

Reply: We have added the synthetic scheme as Figure 2.

- *There are many small errors in English language which should be corrected (e.g. missing articles).*

Reply: The manuscript was significantly rewritten and thoroughly edited before submission. We hope that we have corrected all linguistic and grammar mistakes.

- *Supporting information: Fig. S4/S5: Label * in NMR spectra as d8-THF.*

Reply: We have modified Fig. S4/S5.

- *Supporting information: Fig. S6. FTIR analysis: It would be nice to have the characteristic stretching bands added into the figure as well as in the caption.*

Reply: The assignments of signals are provided in the SI. Moreover, we marked important bands in the IR spectrum in Fig. S7, which should improve clarity.

- *Supporting information: Fig. S7/S8: The labels (a), (b), (c) and so on in the figure caption should be added beside the corresponding powder X-ray diffraction pattern in the figure.*

Reply: The labels have been added.

REVIEWERS' COMMENTS:

Reviewer #1 (Remarks to the Author):

The authors have made a valiant effort to address the issues raised and I am in favour of publication.